# Epigenetic Downregulation of Hsa-miR-193b-3p Increases Cyclin D1 Expression Level and Cell Proliferation in Human Meningiomas

**DOI:** 10.3390/ijms241713483

**Published:** 2023-08-30

**Authors:** Paulina Kober, Beata Joanna Mossakowska, Natalia Rusetska, Szymon Baluszek, Emilia Grecka, Ryszard Konopiński, Ewa Matyja, Artur Oziębło, Tomasz Mandat, Mateusz Bujko

**Affiliations:** 1Department of Molecular and Translational Oncology, Maria Sklodowska-Curie National Research Institute of Oncology, 02-781 Warsaw, Poland; 2Department of Experimental Immunology, Maria Sklodowska-Curie National Research Institute of Oncology, 02-781 Warsaw, Polandryszard.konopinski@pib-nio.pl (R.K.); 3Department of Experimental and Clinical Neuropathology, Mossakowski Medical Research Institute, Polish Academy of Sciences, 02-106 Warsaw, Poland; 4Department of Neurosurgery, Maria Sklodowska-Curie National Research Institute of Oncology, 02-781 Warsaw, Poland

**Keywords:** meningioma, DNA methylation, epigenetics, microRNA, hsa-miR-193b-3p, cyclin D1

## Abstract

Meningiomas are common intracranial tumors in adults. Abnormal microRNA (miRNA) expression plays a role in their pathogenesis. Change in miRNA expression level can be caused by impaired epigenetic regulation of miRNA-encoding genes. We found the genomic region covering the *MIR193B* gene to be DNA hypermethylated in meningiomas based on analysis of genome-wide methylation (HumanMethylation450K Illumina arrays). Hypermethylation of *MIR193B* was also confirmed via bisulfite pyrosequencing. Both hsa-miR-193b-3p and hsa-miR-193b-5p are downregulated in meningiomas. Lower expression of hsa-miR-193b-3p and higher *MIR193B* methylation was observed in World Health Organization (WHO) grade (G) II/III tumors as compared to GI meningiomas. *CCND1* mRNA was identified as a target of hsa-miR-193b-3p as further validated using luciferase reporter assay in IOMM-Lee meningioma cells. IOMM-Lee cells transfected with hsa-miR-193b-3p mimic showed a decreased cyclin D1 level and lower cell viability and proliferation, confirming the suppressive nature of this miRNA. Cyclin D1 protein expression (immunoreactivity) was higher in atypical than in benign meningiomas, accordingly to observations of lower hsa-miR-193b-3p levels in GII tumors. The commonly observed hypermethylation of *MIR193B* in meningiomas apparently contributes to the downregulation of hsa-miR-193b-3p. Since hsa-miR-193b-3p regulates proliferation of meningioma cells through negative regulation of cyclin D1 expression, it seems to be an important tumor suppressor in meningiomas.

## 1. Introduction

Meningiomas are the most common intracranial tumors in adults [1]. They are diagnosed most frequently in persons aged over 65 and significantly more often in women. The histopathological spectrum of meningiomas is diverse, and the vast majority of meningiomas (80%) fulfill the histological criteria of slowly developing benign malignancies, corresponding to World Health Organization (WHO) grade (G) I. WHO GII (mostly atypical meningiomas) and GIII tumors (mostly anaplastic meningiomas) represent approximately 18.3% and 1.6% of all meningiomas, respectively [1]. Patients with the most frequent benign meningiomas usually have favorable prognoses [1]. Surgical treatment is effective in most cases; however, benign meningiomas tend to progress to more aggressive subtypes. WHO GII and GIII tumors have an invasive phenotype with an increased risk of recurrence, invading normal meninges, skull, or brain structures, and shorter survival time [2].

In recent years, significant progress has been made in understanding the molecular basis of meningiomas including the important role of epigenetics [3]. Profiling of DNA methylation contributed to the development of molecular subclassification of meningiomas and shed some light on the role of aberrant methylation pattern in the expression of tumor-related genes [4]. Our study was initiated by the observation of high DNA methylation level at the chromosome 16p13 locus encoding a small cluster of microRNA (miRNA) genes as result of our genome-wide methylation profiling in meningiomas. 

In brief, miRNAs are small (18–24 nucleotides), endogenous, single-stranded non-coding RNA molecules that are involved in negative regulation of gene expression at the post-transcriptional level [5]. Alterations in miRNA expression play an important role in tumorigenesis [5]. Due to their functional role, some miRNA genes are considered tumor suppressors. Lowering their expression level or deleting the miRNA-encoding locus in tumor cells may lead to an overexpression of target genes of oncogene nature. As a consequence, a decline in suppressive miRNA expression contributes to tumorigenesis [6].

MiRNAs are also a source of promising, clinically relevant biomarkers for early cancer detection and prognosing patient outcome. These small molecules can be efficiently detected in various tissue types including biofluids and formalin-fixed tissue, and they can be incorporated in the diagnostic process [7].

Many such miRNAs were found downregulated in human meningiomas [8]. Importantly, the expression level of miRNA in cancer may be reduced due to aberrant DNA methylation at particular genomic loci containing miRNA-encoding genes [9].

This study was aimed to validate the role of DNA hypermethylation of the *MIR193B* gene in meningiomas. Since this gene encodes known onco-suppressive miRNA, we hypothesized that its aberrant DNA methylation may have functional implication for meningioma growth.

## 2. Results

### 2.1. MIR193B/MIR365A Genes Are Hypermethylated in Human Meningiomas

Previous results of DNA methylation profiling with HumanMethylation450K arrays (Illumina) in twenty-four meningiomas (ten benign, eight atypical, and six anaplastic meningiomas) as well as six normal meningeal samples were used to identify differentially methylated regions (DMRs) in meningiomas and normal meninges. Based on quality control, three samples were excluded from the analysis (one WHO GI and one GIII sample as well as one normal sample).

We identified 60 DMRs that met strict criteria of adjusted *p*-value < 0.05, with the number of probes in the DMR ≥ 5 and a Δβ value > |0.3|. The identified DMRs are presented in Figure 1A and in Appendix A. Interestingly, two regions at the genomic miRNA cluster at chr16.p13.12 coding for hsa-miR-193b and hsa-miR-365a microRNAs were identified among the most significant DMRs (according to *p*-value). Both DMRs were notably DNA hypermethylated in meningiomas. One of them included eight differentially methylated probes (DMPs) at *MIR193B*, with a median Δβ value of 0.45, and the other included seven DMPs at *MIR365A* with a median Δβ of 0.4. DMPs annotated to *MIR193B* and *MIR365A* are presented in Figure 1B,C and listed in detail in Appendix A.

### 2.2. DNA Hypermethylation of MIR193B Corresponds to Decreased Expression of Hsa-miR-193b-3p and 193b-5p in Meningiomas

Using a quantitative reverse transcription polymerase chain reaction (qRT-PCR), we determined the expression of mature miRNAs hsa-miR-193b-3p, hsa-miR-193b-5p, hsa-miR-365a-3p, and hsa-miR-365a-5p in meningiomas (*n* = 40) and normal meninges (*n* = 4). We observed significantly lower expression of hsa-miR-193b-3p and hsa-miR-193b-5p in meningiomas, while no significant difference was observed in the expression levels of hsa-miR-365a-3p and hsa-miR-365a-5p. Results are presented in Figure 2. Based on these results, we focused on the role of *MIR193B*.

### 2.3. MIR193B DNA Methylation and Expression of Hsa-miR-193b-3p Diffrentiate Benign and Atypical Meningiomas

We compared *MIR193B* methylation levels (with a PCR amplicon covering DMPs cg04018325 and cg06273075) and the expression of mature miRNAs originated from this gene in meningiomas of different grades. Benign GI meningiomas were compared with atypical GII, and anaplastic GIII meningiomas. Comparing *MIR193B* methylation with pyrosequencing showed a slight, but significant, increase in average methylation levels between GI and GII tumors, while no significant difference was found in GI vs. GIII meningiomas (Figure 3A). The expression of hsa-miR-193b-3p was lower in GII as compared to GI tumors, while no difference was observed in the comparison of GI vs. GIII meningiomas (Figure 3B). No significant differences were observed in the comparison of hsa-miR-193b-5p expression levels among different grade meningiomas, although in our study group it was slightly higher (statistically nonrelevant) in GIII meningiomas compared to benign tumors (Figure 3B).

### 2.4. Hsa-miR-193b-3p Regulates the Expression of CCND1 and Level of Cyclin D1 in Meningioma IOMM-Lee Cells

We used TargetScan (a miRNA target prediction tool) and miRTarBase (a database of experimentally validated miRNA targets) to determine a putative important target of hsa-miR-193b-3p in meningiomas. Based on these two resources, we identified the *CCND1* (cyclin D1) gene as highly probably regulated by this miRNA.

IOMM-Lee meningioma cells were transiently transfected with miRNA mimics, hsa-miR-193b-3p and hsa-miR-193b-5p, as well as control unspecific miRNA. Introducing hsa-miR-193b-3p mimic resulted in decreased cyclin D1 levels in the cells, while no effect of hsa-miR-193b-5p mimic on cyclin D1 expression was observed (Figure 3A).

Direct interaction between hsa-miR-193b-3p and 3′ untranslated region (UTR) of *CCND1* mRNA was verified using luciferase reporter assay. A fragment of 3′UTR of *CCND1* including putative hsa-miR-193b-3p binding motif was identified using TargetScan. This fragment was cloned into pmirGLO plasmid vector at the 3′ region of the firefly luciferase gene. Additionally, this 3′UTR with a modified sequence at the putative hsa-miR-193b-3p binding motif was cloned and used as a mutated 3′UTR negative control. IOMM-Lee cells were transfected with an empty vector, a vector containing hsa-miR-193b-3p binding site, and a vector containing a mutated hsa-miR-193b-3p binding motif. Each of the three variants of the cells was co-transfected with miRNA mimic hsa-miR-193b-3p or unspecific control miRNA mimic. Luminescence was developed 48 h after transfection and detected with a microplate reader. We observed a significant decrease of luminescence in the cells transfected with plasmid containing the putative hsa-miR-193b-3p binding site that was treated with hsa-miR-193b-3p mimic, which confirms the physical interaction between this miRNA and the 3′UTR sequence in a plasmid vector. No luminescence decrease was observed in other variants of the experiment including control cells transfected with the 3′UTR binding site and unspecific miRNA mimic and cells that were transfected with the hsa-miR-193b-3p mimic and the mutated 3′UTR binding site. Results are shown in Figure 4.

### 2.5. Hsa-miR-193b-3p Influences Cell Viability and Proliferation

Using transient transfection of IOMM-Lee meningioma cells with hsa-miR-193b-3p miRNA mimic, we explored the influence of this miRNA on selected cell phenotype features. We determined cell viability with MTT test, cell proliferation with BrdU incorporation-based test, and cell migration using a scratch assay. Each test was performed in two independent biological replicates.

The MTT assay showed lower viability in cells treated with hsa-miR-193b-3p miRNA mimic as compared to negative control miRNA mimic (Figure 5A). Transfection with hsa-miR-193b-3p miRNA mimic also resulted in lower cell proliferation; however, the results in one of the biological replicates of the experiment did not cross the significance threshold (Figure 5B). In turn, we did not find any influence of hsa-miR-193b-3p on cell migration in the scratch assay (Figure 5C).

### 2.6. The Expression of Cyclin D1 Is Higher in High-Grade Than Benign Meningiomas

The cyclin D1 protein level was assessed upon immunohistochemical (IHC) staining of meningioma tissue samples (including twenty-three GI, fifteen GII, and two GIII tumors). For the protein expression analysis, we were not able to include exactly the same samples as we used for the DNA methylation/expression evaluation. Cyclin D1 expression was assessed in fourteen samples from this group (including eleven GI, one GII, and two GIII tumors) and additional tumor samples.

Most tumor samples showed both nuclear and cytoplasmic immunoreactivity, while pure nuclear staining was observed in 9/40 (22.5%) samples. Nuclear and cytoplasmic expression was evaluated separately using the H-index method. We observed significantly higher nuclear cyclin D protein levels in atypical GII meningiomas than benign GI tumors, (*p* = 0.038) (Figure 6A) while no difference was observed in cytoplasmic immunoreactivity. Representative examples of tissue staining are presented in Figure 6B.

## 3. Discussion

The role of epigenetics in pathogenesis of meningiomas has been a subject of research for many years, and the results on DNA methylation especially contributed to current knowledge in that field. Profiling DNA methylation patterns in meningiomas has allowed for specific clinically relevant molecular classification [10,11] and has provided an insight into the biology of these tumors. Aberrant DNA methylation has functional implications, and it changes the expression of particular tumorigenesis-related genes such as *CDKN2A* [12], *TIMP3* [13], *DLC1* [14], or *TERT* [15]. 

The expression of non-coding RNA genes including those encoding miRNA can be epigenetically regulated via DNA methylation as regular protein-coding genes [16,17]. Importantly, aberrant DNA methylation at miRNA genes commonly causes a change in the miRNA expression profile in human cancer [9]. By analyzing the data from our previous DNA methylation profiling in human meningiomas, we found hypermethylation in the miRNA cluster containing *MIR193B* and *MIR365A*. Evaluation of the expression of mature miRNAs that originated from these genes showed that levels of hsa-miR-193b-3p and hsa-miR-193b-5p are lower in meningiomas as compared to normal meninges. Importantly, we found that expression of hsa-miR-193b-3p is lower in atypical meningiomas as compared to benign GI tumors. This observation is in line with previous results from El-Gewely MR et al. who found hsa-miR-193b among miRNAs differentially expressed in GI and GII meningiomas [18]. Moreover, our results show that lower hsa-miR-193b-3p expression corresponds to a slight increase in *MIR193B* DNA methylation level in GII as compared to benign meningiomas. The DNA hypermethylation-related downregulation of miR-193b, which we observe in meningiomas, was previously found in prostate cancer [19,20,21], liposarcoma [22], and esophageal tumors [23].

In general, it appears that hsa-miR-193b-3p is a tumor suppressor [24], in spite of discovering its higher expression in glioma and colorectal cancer [25,26]. Reduced expression levels of hsa-miR-193b were observed in prostate, lung, pancreatic, gastric, and liver cancers as well as in melanoma, where it exerts an inhibitory effect on the proliferation, migration, invasiveness, or metastasis of cancer cells [19,27,28,29,30,31]. In our study, we also observed a suppressive effect of hsa-miR-193b on both cell viability and proliferation in meningioma cells. However, we did not detect its role in cell migration as it was found in pancreatic, breast, and urothelial cancer cells [31,32,33].

Tools for miRNA target prediction, such as TargetScan, which we used in our study, indicate a large number of hsa-miR-193b-3p putative target mRNAs in humans. A few target genes of this miRNA had already been validated using experimental approaches, revealing the mechanism of its suppressive activity [24]. This includes known oncogenes like *KRAS* [34], *ERɑ* [35], *ETS1* [31], and *CCND1* [27,30].

Considering the influence of hsa-miR-193b-3p on the proliferation of IOMM-Lee cells and expression of *CCND1* in meningiomas [36], we focused on experimental verification of the role of this miRNA in regulation of cyclin D1 levels. Cyclin D1 is a key element of the cell cycle progression trigger in the cell which is abnormally expressed in various human cancers [37]. This protein is overexpressed in meningiomas [18], and its expression level correlates with proliferation markers such as Ki67 and PCNA in meningeal tumors [38,39].

Our results on IOMM-Lee meningioma cells confirm that hsa-miR-193b-3p downregulates cyclin D1 levels, and it binds the predicted target site at *CCND1* 3′UTR. Accordingly to the observation of lower expression of hsa-miR-193b-3p in GII than in GI meningiomas, we noticed increased cyclin D1 protein levels in atypical as compared to benign tumors. A similar relationship between cyclin D1 and meningioma grade with a stepwise increase in cyclin D1 expression levels from GI to GIII tumors was also previously reported [39,40]. These results are concordant with the classification of meningiomas in which higher cellular proliferation and mitotic activity are among diagnostic histological hallmarks of atypical and anaplastic meningiomas [2].

Atypical and anaplastic meningiomas represent the definite minority of all meningeal tumors, but treatment is challenging, and the prognoses for the patients are poor [1]. Importantly, benign meningiomas tend to progress over time to more aggressive subtypes. [2]. Our results show the mechanism that partially explains the difference in proliferation potential between benign and aggressive meningiomas. We may assume that *MIR193B* methylation and expression of hsa-miR193b-3p could play a role in the progression of GI to GII tumors; however, it requires further research.

Cell cycle progression is considered a pharmacological target for treatment of meningioma patients. Recently, a few clinical trials evaluating the efficiency of inhibitors of cyclin-dependent kinases (CDKs) have started [41]. Our results support using this class of drugs, especially in more aggressive meningiomas, in which cyclin D1/CDK pathway appears activated. Additionally, the role of hsa-miR193b in cell cycle regulation was observed to contribute in the response of glioblastoma cells to temozolomide [42]. The efficiency of temozolomide in the treatment of meningioma patients was investigated with unsatisfying results, and the mechanism of resistance is unclear [43,44].

In our study, we focused on the functional role of the hypermethylation of *MIR193B*. The analysis of tissue samples was based on a retrospective group of patients. The proportions between histological subtypes do not reflect the true proportions of meningioma histological subtypes in the general population of patients. We recognize the composition of this patient cohort as a study limitation. It allowed for a conclusive comparison of the selected biological features between the subtypes, but we did not examine the possible clinical role of hsa-mir-193b expression in our group of patients. We believe that reliable investigation of its detailed clinical and prognostic relevance would require prospective observation of representative groups of patients. 

In our opinion, the investigation of the role of hsa-miR193b and cyclin D1 expression as possible biomarkers and their role in therapy response provides a promising prospect for further research.

## 4. Materials and Methods

### 4.1. Description of the Patients

Fifty-eight archival formalin-fixed, paraffin embedded (FFPE) meningioma samples from patients who underwent surgical tumor resection at the Maria Sklodowska-Curie National Research Institute of Oncology in Warsaw were used. The study group included 58 patients with meningiomas (24 WHO GI, 22 GII, and 12 GIII). All tissue samples were histologically examined for a review of the diagnosis and selection of a representative tissue section suitable for molecular analysis. Patients’ characteristics are listed in Table 1. Control samples of leptomeninges were harvested during non-oncological neurosurgical procedures while duroplasty was an element of those procedures. RNA and DNA were isolated from FFPE tissue using the RecoverAll™ Total Nucleic Acid Isolation Kit for FFPE (Thermo Fisher Scientific, Waltham, MA, USA) and measured using NanoDrop 2000 (Thermo Fisher Scientific). The samples were stored at −70 °C.

### 4.2. DNA Methylation

Previously generated data on genome-wide methylation in meningiomas obtained using Infinium HumanMethylation450K BeadChip (Illumina, San Diego, CA, USA) were used. This data set included results for 10 benign (GI), 8 atypical (GII), and 6 anaplastic meningiomas (GIII), as well as 6 normal meningeal samples. Data are available at Gene Expression Omnibus (accession GSE241956). Data were analyzed using the minfi Bioconductor package [45]. Methylation probes were filtered (single nucleotide polymorphism (SNP), and probes that failed in at least 25% of samples were excluded). Differentially methylated probes were identified using minfi [45] while DMRs were identified with comb-*p* [46].

The DNA methylation level of the *MIR193B* promoter was determined using bisulfite pyrosequencing as previously performed. One microgram of DNA was bisulfite-treated using the EpiTect kit (Qiagen, Hilden, Germany). A PCR reaction was performed in a volume of 30 μL containing 1× PCR buffer, 2 mM MgCl_2_, 0.25 mM dNTPs, 0.2 μM of each primer, and 0.5 U of FastStart DNA Polymerase (Roche Applied Science, Mannheim, Germany). Cycling conditions were as follows: 94 °C for 3 min, followed by 40 cycles of 30 s at 94 °C, 30 s at 52 °C, and 30 s at 72 °C with a final elongation for 7 min at 72 °C. PCR products were verified on 2% agarose gel, purified using PyroMark Q24 Vacuum Workstation (Qiagen), and analyzed using the PyroMark Q24 System (Qiagen), according to manufacturer’s protocols. The average methylation levels of CpGs within the analyzed sequence were calculated for each sample. The following primers were used: 5′ biotin-labeled forward TTTTTAGTTATGGTGTGGTAAATGT, reverse CCTCTTTTCCCAAAAAATAAATCC for PCR, and CCAAAAAATAAATCCCCAT for sequencing.

### 4.3. Measurement of miRNA Relative Expression Levels

Relative expression levels of mature miRNA were determined using the miScript miRNA PCR System (Qiagen). Total RNA was subjected to reverse transcription using the miScript II RT Kit (Qiagen) while the miScript SYBR^®^ Green PCR Kit (Qiagen) and miScript primer assays were used for PCR amplification, according to manufacturer’s recommendations. The following miScript primer assays were used: MS00031549 (for hsa-miR-193b-3p), MS00008939 (for hsa-miR-193b-5p), MS00031801 (hsa-miR-365a-3p), and MS00041762 (hsa-miR-365a-5p), as well as MS00033740 (for snRNA RNU6B) that served as reference. The specificity of qRT-PCR amplification was confirmed with dissociation curve analysis.

### 4.4. Immunohistochemistry

IHC was performed on 4 μm FFPE tissue sections using the Envision Detection System (DAKO, Glostrup, Denmark). Tissue sections were deparaffinized with xylene and rehydrated in a series of ethanol solutions of decreasing concentrations. Heat-induced epitope retrieval was applied by incubating the samples in Target Retrieval Solution pH 6 (DAKO) in a 96 °C water bath, for 30 min. Cooled slides were treated with a Blocker of Endogenous Peroxidase (DAKO) for 5 min and then incubated with the primary polyclonal antibody (PA5-16777, Thermo Fisher Scientific) in a dilution of 1:100 for 1 h at room temperature. The color reaction was developed with 3,3′-diaminobenzidine tetrahydrochloride (DAKO) as a substrate, and nuclear contrast was achieved via hematoxylin counterstaining. Analysis of immunohistochemical reactivity was performed by calculating the H-score, which combines information on both reaction intensity (scored from 0 to 3) and number of cells with a given intensity. The previously reported formula was used for quantification [47]. Scoring results were analyzed as continuous variables.

### 4.5. In Vitro Cell Line Culture and Transfection with miRNA Mimics

IOMM-Lee cells were purchased from the ATCC collection and cultured in DMEM medium supplemented with 10% FBS and 1% Pen Strep (15140122, Gibco, Waltham, MA, USA). MiScript miRNA Mimics including hsa-miR-193b-3p mimic (MSY0002819, Qiagen), hsa-miR-193b-5p mimic (MSY0004767, Qiagen), and negative control MiScript mimic (Qiagen) were used. A total of 5 × 10^4^ IOMM-Lee cells were seeded per well in culture medium in a 24-well plate and transfected with 50 nM miRNA with 1% (*v*/*v*) HiPerFect Transfection Reagent (Qiagen), according to the manufacturer’s instructions. The next day, the culture medium was changed to fresh basic medium. The cells were harvested or subjected to in vitro tests at 48 h after transfection.

### 4.6. Western Blot

Cells were lysed in ice-cold RIPA buffer, incubated for 30 min at 4 °C, and centrifuged at 12,500 rpm at 4 °C for 20 min. Samples were electrophoresed with SDS-PAGE and transferred to polyvinylidene fluoride membranes (PVDF) (Thermo Fisher Scientific, Waltham, MA, USA). Cyclin D1 was detected with rabbit polyclonal antibody (PA5-16777, Thermo Fisher Scientific) and secondary anti-rabbit antibody conjugated to horseradish peroxidase (HRP) (#7074, Cell Signalling). Anti-lamin A + C antibody (ab108922, Abcam, Cambridge, England) detected with rabbit HRP-conjugated antibody served as the control. SuperSignal West Pico Chemiluminescent Substrate (Thermo Fisher Scientific) and CCD digital imaging system Alliance Mini HD4 (UVItec Limited, Cambridge, United Kingdom) were used for visualization. ImageJ 1.52i software (National Institutes of Health, Bethesda, MD, USA) was used for densitometry.

### 4.7. Luciferase Reporter Gene Assay

The Hsa-miR-193b-3p target site in 3′UTR of *CCND1* was identified using TargetScan [48].

The fragment of *CCND1* 3′UTR containing the putative hsa-miR-193b-3p binding site was cloned into pmirGLO Dual-Luciferase miRNA Target Expression Vector (Promega, Madison, WI, USA), according to the protocol recommended by the plasmid vector supplier. The following sequences of oligonucleotides for cloning were used for wild type fragments of *CCND1* 3′UTR: 5′-AATTCATTTATTGCAGAGGATGTTCATAAGGCCAGTATGATTTATAGCGAT (sense strand) and 5′-CGCTATAAATCATACTGGCCTTATGAACATCCTCTGCAATAAATG (antisense strand). The sequences of 5′-AATTCATTTATTGCAGATGATGTTCATAAGATGAGTATGATTTATAGCGAT (sense strand) and 5′-CGCTATAAATCATACTCATCTTATGAACATCATCTGCAATAAATG (antisense strand) were used for cloning fragments of *CCND1* 3′UTR with the mutated putative hsa-miR-193b-3p binding site. IOMM-Lee cells (2 × 10^4^/well) were seeded onto 96-well plate in 100 µL of medium. The next day, the cells were transfected with 100 ng of plasmid vector (3′UTR-containing or control), using 0.25% (*v*/*v*) lipofectamine 3000 (Invitrogen, Waltham, MA, USA) in 10 µL of DMEM. The cells were subsequently transfected with either hsa-miR-124-3p mimic (YM00471256, Qiagen) or negative control mimic (YM00479902-ADB, Qiagen) in a final concentration of 50 nM using HiPerFect Transfection Reagent (Qiagen). Luciferase activity was measured with the Dual-Glo Luciferase Assay System (Promega) 48 h after transfection.

### 4.8. Functional In Vitro Assays

Cell viability was measured with MTT reagent (Sigma-Aldrich, Saint Louis, MO, USA). Ten microliters of 5 mg/mL MTT stock solution were added to wells, and cells were incubated for 4 h at 37 °C in the cell culture incubator. Next, the medium was carefully removed, and the cells were flooded with 100 μL dimethyl sulfoxide (DMSO), mixed and incubated at 37 °C for 15 min. Absorbance was measured immediately at 540 nm on a microplate reader. 

Cell proliferation was measured with the Cell Proliferation ELISA, BrdU (colorimetric) kit (Sigma-Aldrich), following manufacturer’s protocol. The cells were incubated with BrdU reagent for 6 h at 37 °C in a cell culture incubator. Absorbance was measured at 450 nm and 690 nm using a Victor 3 microplate reader with Wallac 1420 software version 3.00 revision 5 (Perkin Elmer, Wellesley, MA, USA).

Scratch assay (wound healing assay) was used for measuring cell migration. Cells were seeded in 24-well plates at the density of 3 × 10^5^ cells per well. The next day, the cells were transfected with miRNA mimic and cultured for 48 h. Afterwards, the medium was removed, and a scratch was made in each well using a 200 μL pipette tip. Cells were rinsed with phosphate-buffered saline (PBS) and flooded with serum-free medium. Four marked fields were visualized and captured using microscopy (magnification × 40). The same fields were captured after 3, 6, 24, and 30 h incubation. Scratches were measured in 10 places for each image in ImageJ and converted to percent of the size of the scratch relative to time 0.

### 4.9. Statistical Analysis

A two-sided Mann–Whitney U-test was used for analysis of continuous variables. A significance threshold of α = 0.05 was applied. One way ANOVA followed by the Bonferroni test was used for analysis of Western blot and luciferase reporter assay results. A Shapiro–Wilk test was used to verify normal distribution. Data were analyzed and visualized using GraphPad Prism 9.5.1 (GraphPad Software).

## 5. Conclusions

The genomic region at 16.p13.12 covering miRNA-encoding genes *MIR193B* and *MIR365A* is commonly hypermethylated in human meningiomas. Hypermethylation of *MIR193B* is related to downregulation of hsa-miR-193b-3p and hsa-miR-193b-5p in meningiomas. The DNA methylation level of *MIR193B* is higher in atypical WHO GII as compared to benign WHO GI tumors, which corresponds with higher hsa-miR-193b-3p expression in GII than GI meningiomas. Hsa-miR-193b-3p interacts with 3′UTR of *CCND1* mRNA, which results in lowering cyclin D1 expression levels, and influences cell viability and proliferation. Lower expression of cyclin D1 is observed in atypical rather than in benign meningiomas. Our study indicates an onco-suppressive function of hsa-miR-193b-3p and a direct role of DNA hypermethylation of *MIR193B* in the increase of cellular proliferation in meningiomas that relates to overexpression of cyclin D1.

## Figures and Tables

**Figure 1 ijms-24-13483-f001:**
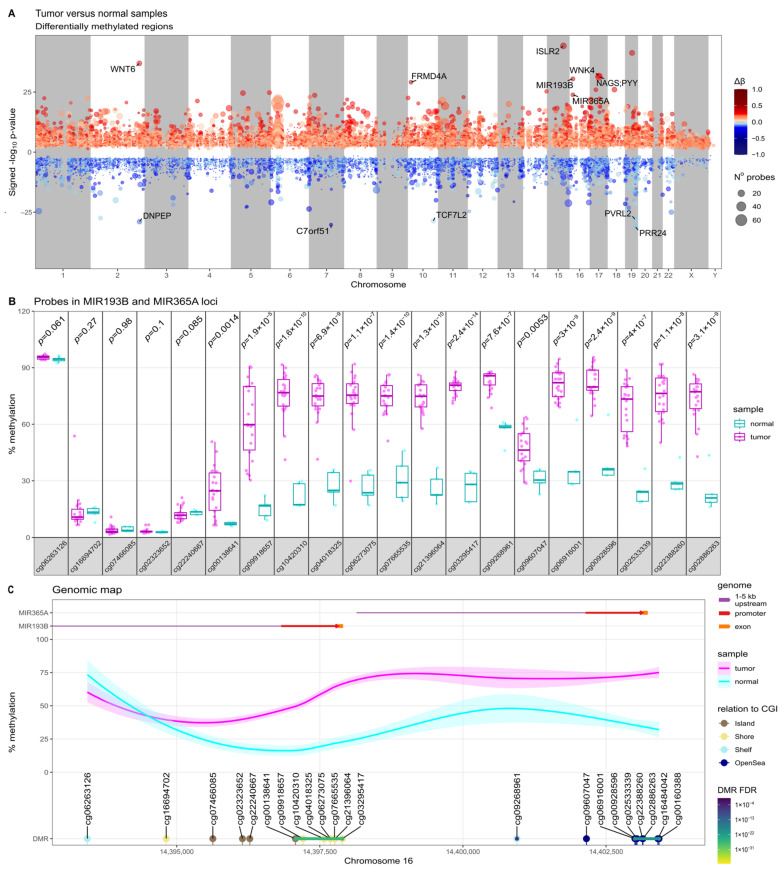
The results of comparison of DNA methylation profile in human meningiomas and normal meningeal samples. (**A**) The genomic map of differentially methylated regions (DMRs); (**B**) comparison of DNA methylation (%) at CpGs covered by HumanMethylation450 K probes located in *MIR193B* and *MIR365A* genes at chr16.p13.12; (**C**) Genomic map of chr16.p13.12 locus presenting the positions of *MIR193B* and *MIR365A* genes, the identified DMRs, and differentially methylated probes (DMPs) as well as DNA methylation level in meningiomas (pink line) and normal samples (blue line). CGI – CpG Island; FDR – false discovery rate.

**Figure 2 ijms-24-13483-f002:**
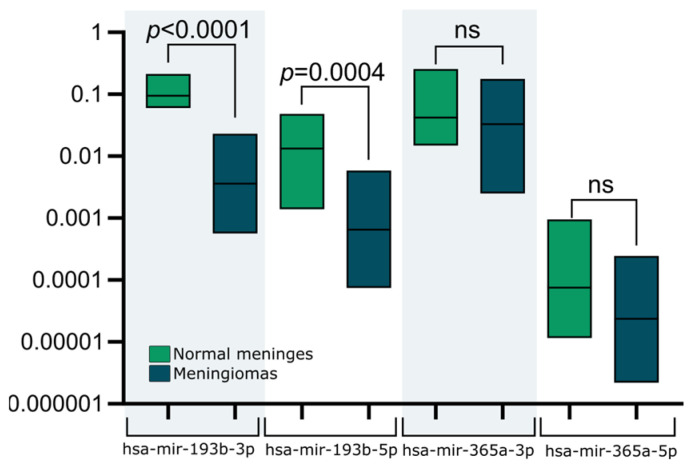
Relative expression level of hsa-miR-193b-3p, hsa-miR-193b-5p, hsa-miR-365a-3p, and hsa-miR-365a-5p in meningiomas (*n* = 58) and normal meninges (*n* = 4). For significant differences between normal meninges and meningiomas *p*-values are shown above the bars, ns—not significant.

**Figure 3 ijms-24-13483-f003:**
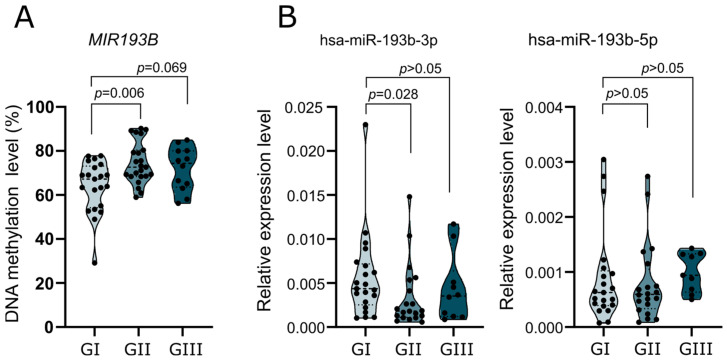
The comparison of benign meningiomas grade (G) I with atypical (GII) and anaplastic (GIII) meningiomas in terms of *MIR193B* DNA methylation and expression levels. (**A**) Comparison of the average methylation level at CpGs covered with bisulfite pyrosequencing assay. (**B**) Comparison of the expression levels of hsa-miR-193b-3p and hsa-miR-193b-5p. Each dot represents a particular tumor sample.

**Figure 4 ijms-24-13483-f004:**
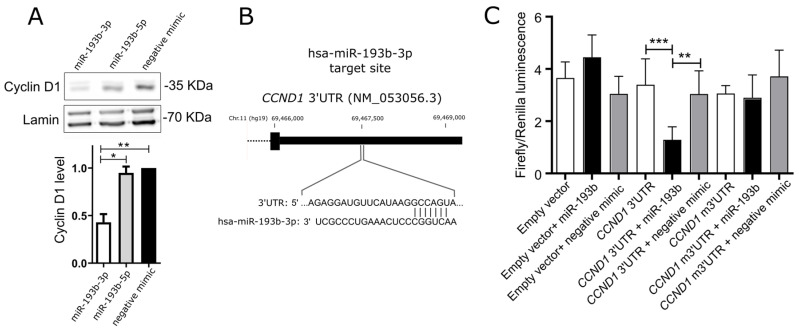
Role of hsa-miR-193b-3p in the regulation of cyclin D1 expression in meningioma IOMM-Lee cell line. (**A**) Decrease of cyclin D1 expression in IOMM-Lee cells transfected with hsa-miR-193b-3p miRNA mimic (western blot membrane and densitometry result). (**B**) Location of the predicted hsa-miR-193b-3p target site in *CCND1* 3′ untranslated region (UTR). (**C**) The results of luciferase reporter assays verifying the interaction between the fragment of 3′UTR of *CCND1* (*CCND1* 3′UTR) and hsa-miR-193b-3p mimic (miR-193b). pmirGLO plasmid without any insert (empty vector) as well as pmirGLO plasmid with mutated sequence of 3′UTR of *CCND1* target site (*CCND1* m3′UTR) were used as controls. Asterisks indicate the statistical significance of differences (* *p* < 0.05, ** *p* < 0.01, *** *p* < 0.001).

**Figure 5 ijms-24-13483-f005:**
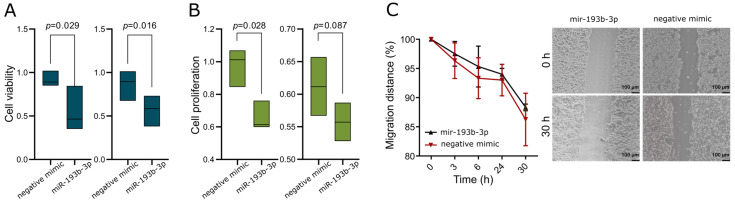
Suppressive role of hsa-miR-193b-3p in IOMM-Lee meningioma cells. (**A**) Reduction of viability in cells transfected with hsa-miR-193b-3p mimic (MTT assay). Two independent replicates of the experiment are presented. (**B**) Lowered proliferation of cells transfected with hsa-miR-193b-3p mimic (BrdU incorporation-based test). Two independent replicates of the experiment are presented. (**C**) The results of scratch assay show no difference in the migration of cells transfected with hsa-miR-193b-3p or negative control mimic. Images were taken with the use of Olympus CKX53 microscope with magnification × 40.

**Figure 6 ijms-24-13483-f006:**
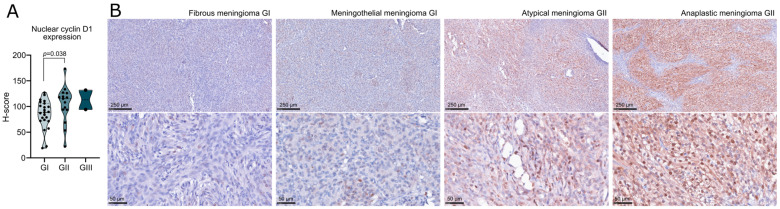
Cyclin D1 protein expression in meningiomas. (**A**) Comparison of nuclear immunoreactivity against cyclin D1 in benign and atypical meningiomas, quantified with H-score. Each dot represents a particular tumor sample. (**B**) Representative examples of immunohistochemical staining.

**Table 1 ijms-24-13483-t001:** Characteristics of the patients.

Clinical Feature	
Number of patients	58
Females	37/58 (63.8%)
Males	21/58 (36.2%)
Age at surgery (years; median (range))	60 (32–86)
Meningioma subtype	
WHO grade I	24/58 (41.4%)
Meningothelial	15 (25.9%)
Fibrous	5 (8.6%)
Transitional	4 (6.9%)
WHO grade II, atypical	22/58 (37.9%)
WHO grade III, anaplastic	12/58 (20.7%)

## Data Availability

The HM450K data are available on Gene Expression Omnibus (accession GSE241956).

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
