# Peer review of "Epigenetic Downregulation of Hsa-miR-193b-3p Increases Cyclin D1 Expression Level and Cell Proliferation in Human Meningiomas"

_ijms, 2023, doi:10.3390/ijms241713483_

Round 1

Reviewer 1 Report

I have read this paper and I have got some comments:

1. In the abstract and main section, all abbreviations have to be explained at the first used.

2. introduction is a good płace to explain why miRNAs can be good Molecular markers; why only one miRNA was chowem for futher analysis.

3. In the statistical analysis Please add the name of test used to analyzed normalility of the date distribution.

4. Please add a subsection of the statistical analysis - sample sieze calculation.

5. Please add a Western blot Images in the main text (a representativeexample).

6. The figures and tables have to self-explained. Please revise it.

7. Please describe how the specify of PCR was confirmed.

8. Wound healing test should be performed.

9. The authors should analyzed the changes in the expression profile of the apoptotic and antiapoptotic genes in order to answear the question about proliferation.

10. References and whole paper have to be prepared according to the Journal’s requirements.

11. In the discussion section Please add more clinical aspects of your work.

12. Please add as the last paragraph in the discussion section limitations, strenghts and future perspectives.

13. Authors should prepare a predective analysis between analyzed miRNA ang genes. It allow to show the relationship at the observed chages; and discuss it.

14. Conclusions have to be correspond With the study aimed and obtained results.

Some minor changes is recommended.

Reviewer 2 Report

In this study, the authors analyzed the epigenetic silencing of the putative tumor suppressor gene MIR193B in human meningioma samples by employing a combination of Illumina arrays and bisulfite pyrosequencing. The expression of one of MIR193B gene products (i.e., the hsa-miR193b-3p species) was found to be downregulated in higher-grade meningiomas compared to benign meningiomas. Moreover, the 3’UTR of CCND1 mRNA (i.e., the cyclin D1 transcript) was identified as a target of has-miR193b-3p, which was further confirmed by the authors in a bioluminescent reporter assay. Lastly, IOMM-Lee human meningioma cells transfected with a hsa-miR193b-3p mimic showed decreased cyclin D1 levels and proliferation as measured by Western blot and BrdU incorporation. All these findings appear to implicate the hsa-miR193b-3p microRNA species as a novel tumor suppressor mechanism in the progression of higher grade meningiomas. While these findings are quite interesting and generally based on a sound methodology, potentially leading to the discovery of a new actionable target in meningioma, I have a couple of comments for the authors as follows:

1.    Cyclin D1 appears to play a more significant role as a driver in grade II (atypical) compared to grade III (anaplastic) meningiomas. The latter tumors are further mutated and most probably fed by a number of additional independent drivers in related pathways. This is in line with the observation made by the authors that the epigenetic downregulation of the cyclin D1 regulator hsa-miR193b-3p represents a more significant event in grade II compared to grade III meningiomas. For this reason, it would have made more sense to validate the hypothesized effects of the hsa-miR193b-3p mimic in a meningioma cell line that is derived from a grade II meningioma as opposed to IOMM-Lee which is a cell line established from a grade III meningioma. If WHOII meningioma cell lines are available, I recommend the authors to repeat their in vitro studies with these for more robust effects.  

2.  Regarding the in vitro studies chosen by the authors to validate their hypothesis, I recommend a refinement of their methodology to better reflect the biologic activity of cyclin D downregulation. For instance, the MTT assay may not be an appropriate one for this scope since this assay, although marketed as a ‘viability’ assay, is more of a mitochondrial metabolic assay. I would not recommend the MTT assay for studying either cell death or cell proliferation. In the specific context of the present study, a more appropriate assay to study the effect of the hsa-miR193b-3p mimic would have been a cell cycle analysis by flow cytometry. It would be important to measure the percentage of cells arrested in G1 after hsa-miR193b-3p introduction, which is an expected effect that directly reflects the downregulation of cyclin D1 by this microRNA species. In my opinion, a cell-cycle analysis by flow cytometry would be significantly more informative than the MTT assay in this context and a better companion for the BrdU incorporation analysis.    

Minor comment: the sentence from line 147 that starts with “No fluorescence decrease was observed…” should read “No luminescence decrease was observed…”

Round 2

Reviewer 1 Report

none

Reviewer 2 Report

I thank the authors for their responses. My recommendation for conducting a cell-cycle analysis by flow cytometry with the hsa-miR193b-3p mimic still stands for some very obvious reasons. This type of analysis is very easy to execute in terms of both effort and time investment. It would also be quite informative in terms of visualizing the potential effect of this miR mimic on pushing at least some of the IOMM-Lee cells into cell cycle arrest (specifically G1 arrest). Obviously, this could have further therapeutic implications if done comparatively side-by-side or in combination with palbociclib (i.e., an already-marketed CDK4/6 inhibitor) at clinically relevant concentrations. Obviously, such data would further and greatly benefit the potential reader of this paper. However, if a FACS analyzer (which otherwise is an extremely common piece of equipment) is not available to the authors, then this type of analysis is clearly not possible.